# A Novel Emergency Braking Control Strategy for Dual-Motor Electric Drive Tracked Vehicles Based on Regenerative Braking

**Zhaomeng Chen** [1] , **Xiaojun Zhou** [1,*]**, Zhe Wang** [1]**, Yaoheng Li** [2] **and Bo Hu** [3]

[1]   State Key Laboratory of Fluid Power and Mechatronic Systems, Zhejiang University, Hangzhou 310027, China; chernzm@zju.edu.cn (Z.C.); wzhe@zju.edu.cn (Z.W.)
[2]   China North Vehicle Research Institute, Beijing 100072, China; 13121441@bjtu.edu.cn
[3]   United Automotive Electronic Systems Co., Ltd., Shanghai 201206, China; bo.hu@uaes.com
*   Correspondence: me_zhouxj@zju.edu.cn; Tel.: +86-0571-8795-2516

**Abstract:** Dual-motor electric drive tracked vehicles (DDTVs) have drawn much attention in the trends of hybridization and electrification for tracked vehicles. Their transmission chains differ significantly from the traditional ones. Due to the complication and slug of a traditional tracked vehicle braking system, as well as the difference of track-ground with tire-road, research of antilock braking control of tracked vehicles is rather lacking. With the application of permanent magnet synchronous motors (PMSMs), applying an advanced braking control strategy becomes practical. This paper develops a novel emergency braking control strategy using a sliding mode slip ratio controller and a rule-based braking torque allocating method. Simulations are conducted under various track-ground conditions for comparing the control performance of the proposed strategy with three other strategies including the full braking strategy, traditional antilock braking strategy, as well as sliding mode slip ratio strategy without the use of motors. For an initial speed of 80 km/h, simulation results show that the proposed control strategy performs the best among all strategies mentioned above. Several hardware-in-the-loop (HIL) experiments are conducted under the same track-ground conditions as the ones in the simulations. The experiment results verified the validity of the proposed emergency braking control strategy.

**Keywords:** dual-motor electric drive tracked vehicles; emergency braking control strategy; sliding mode control; regenerative braking

## 1. Introduction

With the increasing emphasis on environmental protection, automobile technology tends to develop in the direction of hybridization and electrification. Dual-motor electric drive tracked vehicles (DDTVs) have drawn much attention in the research of tracked vehicles due to their high transmission efficiency, low fuel consumption, silent driving performance, and easy maintenance [1–6]. What is more, the introduction of two motors with fast dynamic response makes it feasible to control the rotational speed and torque of drive wheels accurately, rapidly, and independently. These characteristics of DDTVs set the stage for preventing the locking of drive wheels and tracks when emergency braking occurs, making full use of the road adhesion ability and improving the controllability of the vehicle under safety-critical driving situations.

Most of the traditional antilock braking systems (ABS) simply rely on high-frequency switching of solenoid valves in hydraulic or pneumatic braking circuits and widely applied in various kinds of vehicles. However, this kind of control strategy may face decays of control effects when the adhesion

coefficient of real roads is different from the one used to calibrate the ABS. Due to the working principle of the traditional ABS, the slip ratio may oscillate greatly in the process of braking, thus make it impossible to make full use of road adhesion. To overcome these disadvantages of traditional ABS, researchers started to seek direct ways to control the slip ratio of vehicles during emergency braking [7] and found that applying a sliding mode wheel-slip controller on a traditional ABS can improve the braking effect vastly whether the mechanical braking system is hydraulic [8–10] or pneumatic [11].

Electric vehicles (EVs) and hybrid electric vehicles (HEVs) have the ability of regenerative braking compared with diesel and petrol vehicles, which makes the emergency braking control strategies of EVs and HEVs different from traditional ways and results in some progress in different aspects. Currently, studies of emergency braking of electric drive vehicles are mostly concentrated on wheeled vehicles, and they can be divided into three main technical routes. The first route is to rely on the mechanical braking system as much as possible. In these control methods, regenerative braking works just in mild braking to regenerate energy and will be reduced when indicators like slip ratio or brake pedal travel exceed predefined criteria. Thus, the mechanical braking system and its antilock braking strategy take control of the vehicle. This control strategy can guarantee braking performance and braking reliability, as well as maintain the traditional mechanical braking structure, which helps to reduce the difficulties of theoretic analysis and application [12,13]. The transient switching progress between mild and emergency braking of these methods is also studied to enhance the performance [14]. The second route aims to regenerate more energy while maintaining vehicle stability during emergency braking. Although different methods such as sliding mode [15,16], phase plane theory [17], or improved linear quadratic Gaussian control [18] are used to derive the demanded braking force, the main strategy is making the motors provide as much braking force as possible, and the mechanical braking system follows the difference of demanded braking force and regenerative braking force [15–19]. However, the dynamic characteristics of these vehicles are not taken into consideration seriously in the above methods. Hence, we have the third route, which takes advantage of the motor's rapid torque response ability. The basic principle of these methods is to use mechanical braking to provide a rather steady and large amount of braking force and let regenerative braking force compensate the error of demanded and mechanical braking force, no matter whether improving the traditional ABS with regenerative braking [20,21], braking allocation based on tire-road recognition [22,23], or using an advanced braking controller based on sliding mode, a low-pass filter, or phase plane theory [24–27].

As for tracked vehicles, studies on emergency braking are relatively rare. A few accessible ones related to combined braking systems are mostly applying hydraulic-mechanical braking systems because hydraulic retarders are widely used in modern high-speed tracked vehicles [28]. Research on traction control of tracked vehicles using the sliding mode method has been conducted long before [29], but since ABS is rarely used on tracked vehicles, the response of a typical hydraulic system is normally slow, and the braking force of the hydraulic retarder is a function of speed, which cannot be controlled actively [28,30]. Braking stably on a low adhesion surface is comparatively difficult for a traditional tracked vehicle. With the development of DDTVs, cooperative control of motors, hydraulic retarders, and mechanical brakes may leads to a better performance of stability, but current studies are still mainly focused on how to generate enough braking force while regenerating more energy [30–32].

Thus, this paper comes up with a novel emergency braking control strategy for dual-motor electric drive tracked vehicles based on regenerative braking to improve braking stability. The braking performances have been improved significantly adopting this strategy compared with traditional ways. The organization of this paper is as follows. In Section 2, the structure of a DDTV and the models related to its braking process are introduced. In Section 3, the design of the braking controller, which includes a slip ratio controller and a braking force allocation method is performed. MATLAB/Simulink simulations of the proposed controller with a typical emergency braking process, as well as some other control strategies are shown and analyzed in Section 4. Section 5 introduces a series of controller-driver-in-the-loop experiments based on a dSPACE SCALEXIO platform. Finally, conclusions are given in Section 6. The nomenclature is listed in Appendix A.

## 2. Structure Overview and Modeling of a DDTV

### 2.1. Structures of a DDTV

The hybrid powertrain system of a DDTV in this case can be classified as a series HEV system, and its schematic structure is shown in Figure 1. Two tracks are attached to the corresponding drive wheels, which are connected to half-drive-shafts through wheel side reducers. The half-drive-shafts are driven by two permanent magnet synchronous motors (PMSMs) coupled by a coupling mechanism, which could realize renewable power mechanical recirculation in small radius steering and reduce the demanded rated power of the PMSM [33]. Similar to the integrated transmission devices used in normal high-speed tracked vehicles, the coupling mechanism consists of several planetary gear sets for different driving situations, like zero radius steering or small radius steering; however, for normal driving situations, it acts just like a reducer that decouples the operating of dual motors. Thus, drive wheels on each side are separately driven by these motors when running and braking in normal use. There is also a hydraulic retarder and a mechanical brake attached to each half-drive-shaft for braking. The DDTV has two power sources, which are a combustion engine-generator set (CEGS) and an energy storage unit (comprised of battery packs and supercapacitors). The transmission control unit (TCU) is used to analyze the driver inputs and control the engine control unit (ECU), the generator control unit (GCU, which consist of a DSP and an AC/DC converter), the DC/DC converter, the motor control unit (MCU, which consist of a DSP and two DC/AC converters for dual PMSMs), and the braking system (including hydraulic retarders and hydraulic mechanical brakes). All these controllers are communicated through a high-speed CAN 2.0 network. This kind of structure provides convenience for coordinated control of motors, hydraulic retarders, and mechanical brakes when emergency braking occurs.

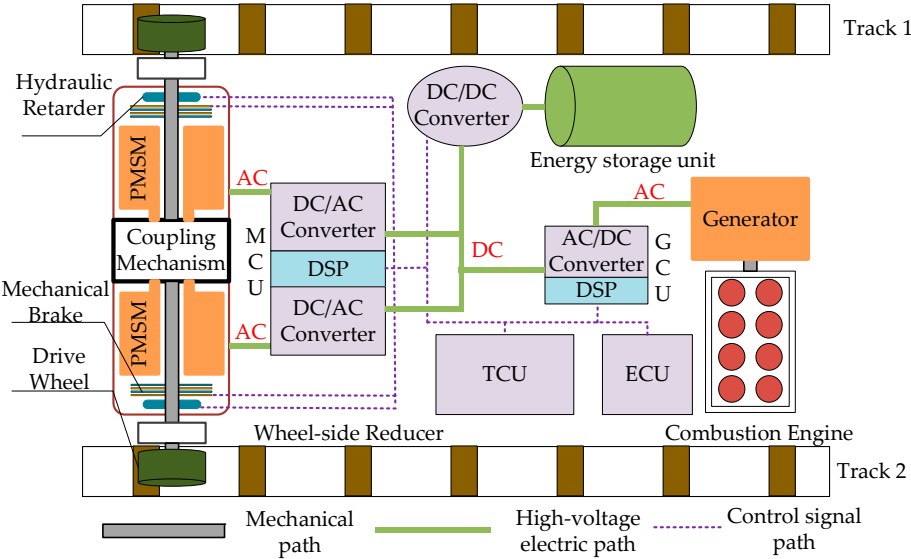

**Figure 1.** Schematic diagram of a dual-motor electric drive tracked vehicle (DDTV) structure. MCU, motor control unit; TCU, transmission control unit; ECU, engine control unit.

### 2.2. Longitudinal Dynamics of a DDTV

As the walking structure of a tracked vehicle and a wheeled vehicle varies greatly, a suitable longitudinal dynamic model should be built appropriately for a DDTV. There are three main components including the vehicle body, the tracks, and the wheels, as shown in Figure 2, so that several freedoms of motion should be taken into consideration, and they are the rolling of the drive wheel, the longitudinal motion of the vehicle body, the rotational motion of the track, and the rolling of the road wheels, idler, and support rollers, respectively. Assuming that the track is uniform and soft and there is no slip or

slide between the track and drive wheel, or the track and road wheels, the idler, or support rollers, a semi-vehicle longitudinal dynamic model in the braking situation can be established as follows.

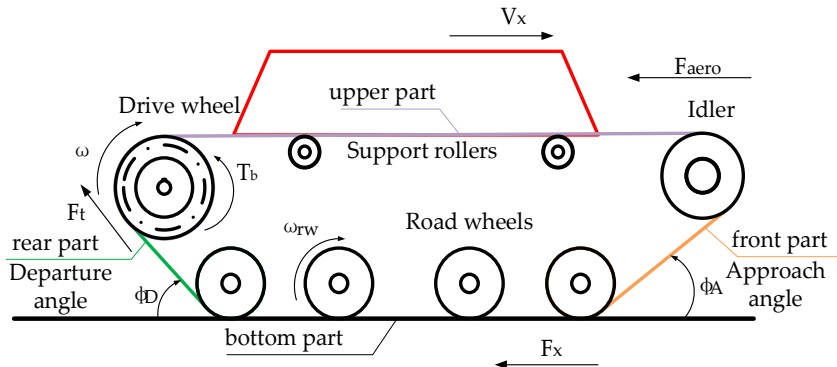

**Figure 2.** Schematic diagram of force and torques acting on a DDTV when braking.

Selecting the forward direction of the vehicle and the corresponding rotational direction of the drive wheel as the positive direction, the dynamics of the longitudinal motion of the vehicle body and rolling of the drive wheel when braking can be denoted as:

$$\begin{cases} F_x + F_{aero} + M\dot{v}_x = 0 \\ J\dot{\omega} - F_t r + T_b = 0 \end{cases} \, , \tag{1}$$

where $F_x$ is the track-ground resultant force of one track, $F_{aero}$ is half of the air drag, $M$ is half of the DDTV's total mass, $v_x$ is the longitudinal velocity of the vehicle, $J$ is the rotational inertia of the drive wheel and transmission system attached to it, $\omega$ is the rotational speed of the drive wheel, $F_t$ is the force applied on the drive wheel by the track, $r$ is the radius of the drive wheel, and $T_b$ is the braking torque applied on the drive wheel by the transmission system.

The track is a component that turns the rotation of the drive wheel into the motion of the vehicle and causes the rotation of the road wheels, idler, and support rollers. Since the mass of the track and the rotational inertia of the attached wheels are far more than an amount that can be ignored and the speeds of each part of the track are not the same due to its irregular shape, a method of calculating the equivalent rotational inertia (ERI) with respect to the rotational axis of the drive wheel based on energy conservation theory can be used to simplify the dynamics of tracks and wheels attached to them [34]. The ERI of the wheels and track can be expressed as follows according to [35]:

$$J_{eri}^{rw} = n_{rw} J_{rw} \frac{r^2}{r_{rw}^2} \, , \tag{2}$$

$$J_{eri}^{sr} = n_{sr} J_{sr} \frac{r^2}{r_{sr}^2} \, , \tag{3}$$

$$J_{eri}^{I} = J_I \frac{r^2}{r_I^2} \, , \tag{4}$$

where $J_{rw}$, $J_{sr}$, and $J_I$ are the rotational inertia of the road wheel, support roller, and idler, respectively, and $n_{rw}$, $r_{rw}$, $n_{sr}$, $r_{sr}$, and $r_I$ are the quantity and the radius of them.

The track can be separated into four parts, which are the front part, rear part, upper part, and bottom part, respectively, as shown in Figure 2, and the ERI of them can be expressed as follows according to [34,35]:

$$\begin{cases} J_{eri}^{t\_f} = 2m_{t\_f} r^2 (1 - \cos \varphi_A) \\ J_{eri}^{t\_r} = 2m_{t\_r} r^2 (1 - \cos \varphi_D) \\ J_{eri}^{t\_u} = 4m_{t\_u} r^2 \end{cases} \, , \tag{5}$$

In Equation (5), $m_{t\_f}$, $m_{t\_r}$, and $m_{t\_u}$ are the masses of the first three parts of the track, and the ERI of the bottom is zero due to the speed of it when the slip of the track and ground is ignored. However, the slip of them can be large when emergency braking occurs, and the speed of every part of the track is different from the ideal situation used in [35], so the ERI of the track should be reconsidered with the slip ratio $\lambda$, which can be defined as follows when braking [36]:

$$\lambda = \frac{v_x - \omega r}{v_x} \ ,$$

(6)

Hence, Equation (5) can be amended as (detailed in Appendix B):

$$\begin{cases} J_{eri}^{t\_f} = m_{t\_f}r^2\left[1 + \frac{1}{(1-\lambda)}^2 - \frac{2cos\varphi_A}{(1-\lambda)}\right] \\ J_{eri}^{t\_r} = m_{t\_r}r^2\left[1 + \frac{1}{(1-\lambda)}^2 - \frac{2cos\varphi_D}{(1-\lambda)}\right] \\ J_{eri}^{t\_u} = m_{t\_u}r^2\left(\frac{2-\lambda}{1-\lambda}\right)^2 \\ J_{eri}^{t\_b} = m_{t\_b}r^2\frac{\lambda}{1-\lambda}^2 \end{cases} ,$$

(7)

According to Equations (1)–(4) and (7), the longitudinal dynamics of the DDTV in this case can be denoted as:

$$\begin{cases} F_x + F_{aero} + M\dot{v}_x = 0 \\ J_{eri}\dot{\omega} - F_x r + T_b = 0 \end{cases} ,$$

(8)

where the rotational inertia $J_{eri}$ is achieved by summing all the ERIs reported in (2)–(4) and (7).

## 2.3. Models of the Track-Ground

There are many mature tire-road models like the Burckhardt tire model [16] and magic formula tire model [25] for wheeled vehicles; however, the structure of the track and the contact form of the track-ground differ greatly from the tire-road [25,37], so there are no suitable analytical models for it. Therefore, the best way to conform to the reality is to analyze and simulate with an experimental model. Experimental track-ground look-up-table models can be obtained through [36] and actual tests. Some typical track-ground conditions are shown in Figure 3.

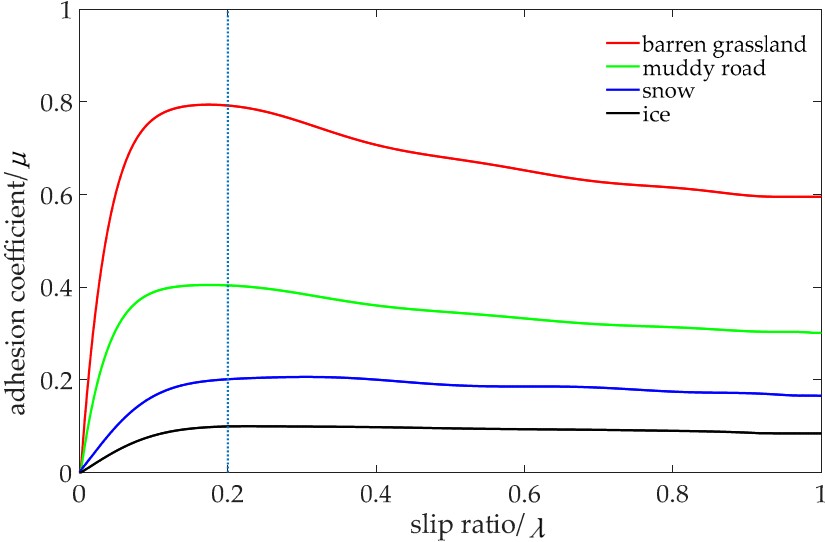

**Figure 3.** Schematic diagram of four typical track-ground conditions, where the adhesion coefficient can be indicated as $\mu$ and slip ration can be indicated as $\lambda$.

### 2.4. Model of the PMSM

The characteristics we focused on of the PMSM in this case were its torque output capacity and response ability. Instead of building a complex model with its internal physical processes, a simple model that can reflect the external characteristic and dynamic behavior is more appropriate. The model of the PMSM used in this case can be expressed as:

$$T_{mo} = max\left(\frac{T_{brdem}}{i_c i_r}, -T_{mo\_max\_n}\right)\frac{k_1}{\tau_1 s + 1} , \tag{9}$$

where $T_{mo}$ is the motor torque, $T_{brdem}$ is the demanded torque of regenerative braking converted to the drive wheel, $i_c$ is the transmission ratio of the coupling mechanism, $i_r$ is the transmission ratio of the wheel-side reducer, and $T_{mo\_max\_n}$ is the maximum torque the motor can output at the current speed according to its torque-speed curve; $k_1$ and $\tau_1$ can be derived according to the experimental data.

### 2.5. Model of the Mechanical Brake

Similar to the modeling of the PMSM, an experimental model of the hydraulic mechanical brake used in this case can be expressed as:

$$T_{me} = \frac{T_{bmdem}}{i_r} \frac{k_2}{\tau_2 s + 1} , \tag{10}$$

where $T_{me}$ is the torque of the mechanical brake, $T_{bmdem}$ is the demanded torque of mechanical braking converted to the drive wheel, and $k_2$ and $\tau_2$ can be derived according to the experimental data.

### 2.6. Model of the Hydraulic Retarder

Hydraulic retarders are widely used in high-speed tracked vehicles to help reduce the speed when the vehicle is braking at a high speed [28,30,33]. However, their torque is not tunable and is a function of its rotational speed for a certain one. The experimental model of the one used in this case can be expressed as:

$$\begin{cases} T_{hr} = T_{hr\_n}\frac{k_3}{\tau_3 s + 1} \\ T_{bhr} = i_r T_{hr} \end{cases} , \tag{11}$$

where $T_{hr}$ is the real torque of the hydraulic retarder, $T_{hr\_n}$ is the torque that the hydraulic retarder can output when the rotational speed of it is at a value of $n$, $k_3$ and $\tau_3$ can be derived according to the experimental data for testing its dynamic response, and $T_{bhr}$ is the torque of it converted to the drive wheel.

### 2.7. Model of the Driver

According to the national military standard of China for electric drive tracked vehicles, the maximum deceleration must be higher than 5 m/s$^2$, and this index is usually set as −5.5 m/s$^2$ [32]. Therefore, the travel of the brake pedal can be interpreted as a demand deceleration. The pedal travel is usually considered as meaningful after a small idle motion, and full braking is deemed to be needed when nearing the terminal. Thus, the driver model can be expressed as:

$$a_{bdem} = \begin{cases} 0 & 0 \leq \beta \leq 5\% \\ -5.5\frac{(\beta-5\%)}{90\%} & 5\% \leq \beta \leq 95\% \\ -5.5 & \beta \geq 95\% \end{cases} \tag{12}$$

where $a_{bdem}$ is the demanded deceleration and $\beta$ is the brake pedal travel.

### 3. Emergency Braking Control Strategy Design

The braking control system mainly contains controller selecting rules, braking control strategies, braking actuators, and vehicle dynamics. The schematic diagram of it is shown in Figure 4.

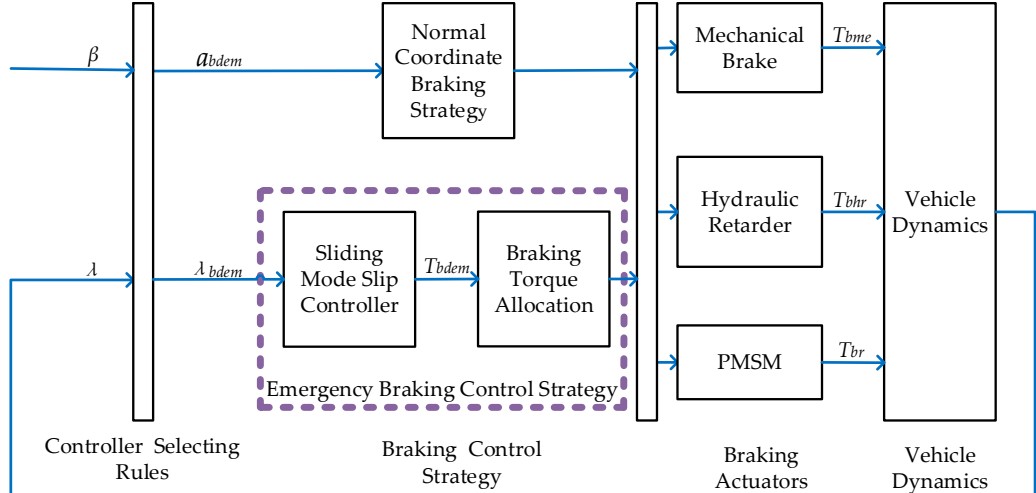

**Figure 4.** Schematic diagram of the braking system.

The first thing that should be clarified is when the emergency braking control strategy needs to be triggered. Without any doubt, when the brake pedal travel exceeds a certain value, for example 95%, as shown in Equation (12), this means the driver wants the vehicle to stop as soon as possible. However, when driving on low adhesion grounds, it is impossible for the vehicle to achieve a deceleration exceeding the ability that the track-ground condition can provide. Thus, when the slip ratio goes higher than a threshold, the emergency braking control strategy needs to be triggered. It is shown in Figure 3 that a $\mu - \lambda$ curve rises fast with the increase of the slip ratio from zero and turns to a flat growth as $\lambda$ approaches 0.15, then when the slip ration goes over 0.2, the adhesion coefficient starts to fall with the growth of $\lambda$, although the falling speed is related to a certain value of $\lambda$ and what the track-ground condition is. Hence, the threshold of the slip ratio can be set as 0.15.

The emergency braking control strategy proposed in this article consists of a sliding mode slip controller and a rule-based braking force allocation method. When the emergency braking control strategy is triggered, the sliding mode slip controller calculates the total braking torque demanded converted to the drive wheel according to the current vehicle status, and then, the braking torque allocation method divides the total demanded braking torque into three parts separately for the mechanical brake, the hydraulic retarder, and the PMSM according to the rules and vehicle status.

*3.1. Sliding Mode Slip Ratio Controller*

According to Equation (6), the derivation of the slip ratio with respect to time is:

$$\dot{\lambda} = \frac{\dot{v}_x \omega r - \dot{\omega} r v_x}{v_x^2} \tag{13}$$

Substitute $\omega$ as a function of $\lambda$ from Equation (6) and $\dot{\omega}$ defined in Equation (8) into Equation (13); we can get that:

$$\dot{\lambda} = f - \frac{(1-\lambda)}{\omega J_{eri}} T_b \tag{14}$$

where:

$$f = \frac{(1-\lambda)^2 J_{eri} \dot{v}_x + (1-\lambda) F_x r^2}{\omega r J_{eri}} \tag{15}$$

Here, $f$ is a function of vehicle deceleration $\dot{v}_x$, drive wheel angular velocity $\omega$, and track-ground resultant force $F_x$. Since $\dot{v}_x$ and $\omega$ are directly measured by the accelerometer and encoder, the errors of them are relatively small, so the error of estimating $f$ is mainly the result of the error estimating $F_x$. Assuming that the estimation of $F_x$ is bounded, then the error of $f$ can be expressed as:

$$\widetilde{f} = \hat{f} - f = \frac{(1-\lambda)}{\omega J_{eri}}\left(\hat{F}_x - F_x\right)$$

(16)

and its boundary can be represented as:

$$F_d = \frac{(1-\lambda)r}{\omega J_{eri}}\widetilde{F}_{x,max}$$

(17)

where $\widetilde{F}_{x,max}$ is the maximum error of estimating $F_x$.

A zero-order sliding surface can be chosen to track the demanded slip ratio [8,13,29]; the sliding surface and its derivation with respect to time are:

$$\begin{cases} s_s = \lambda - \lambda_d \\ \dot{s}_s = \dot{\lambda} - \dot{\lambda}_d \end{cases}$$

(18)

Although the $\mu - \lambda$ curves vary widely according to different track-ground conditions, it is fortunate that the adhesion coefficient always achieves its largest value when the slip ratio nears the point of 0.2, as shown in Figure 3, so the demanded slip ratio can be set as 0.2 for convenience, no matter at what track-ground conditions. Thus, the derivation of the sliding surface equals that of the slip ratio.

In order to converge as quickly as possible when $s$ is large and reduce chattering when $s$ is small, a sliding mode robust control method using the exponential approaching law is suitable for a plant with bounded disturbance [38]. Thus, regarding the error of estimating $f$ as a disturbance, the derivation of the sliding surface can be rewritten as:

$$\dot{s}_s = -\varepsilon sign(s_s) - ks_s = \hat{f} - \frac{(1-\lambda)}{\omega J_{eri}}T_b - \widetilde{f}, \varepsilon > 0, k > 0$$

(19)

where $sign(s_s)$ is a sign function of $s_s$, $k$ is suggested to be designed relatively large to realize a high approaching speed, and $\varepsilon$ is suggested to be designed relatively small to reduce the chattering near the sliding surface. According to Equations (15), (17), and (19), the total demanded braking torque converted to the drive wheel can be derived as (detailed in Appendix C):

$$T_{bdem} = r\hat{F}_x + \frac{1-\lambda}{r}J_{eri}\dot{v}_x + \frac{\omega J_{eri}}{1-\lambda}[(\varepsilon + F_d)sign(s_s) + ks_s]$$

(20)

### 3.2. Rule-Based Braking Torque Allocating Method

There are three independent braking actuators for each half-shaft, and how to allocate the demanded braking torque derived in Equation (20) to them is the key to better follow the change of it and get a better braking performance.

These three braking actuators have their own advantages and drawbacks in different aspects. The mechanical brake can offer a large amount of braking torque, but its dynamic response is quite poor. The hydraulic retarder has a better response than the mechanical brake and can provide a stable braking torque in a rather high revolution speed, but its braking torque is nonadjustable at a certain revolution speed; what is more, its braking torque declines dramatically with the reduction of speed when the half-shaft rotates under a certain speed. As for the PMSM, its torque response is the best among these actuators and even can provide a positive torque in the braking process, which can help to improve the system response. However, according to its model, the maximum torque of PMSM is related to its revolution speed, so the maximum motor torque declines when exceeding the motor's

rated speed, even though the rated torque of it is already much smaller than the mechanical brake can provide; what is more, the accuracy of its torque control and the efficiency of the motor drive system also become worse when the revolution speed of the half-shaft is too low.

Therefore, several basic rules can be set according to the above characteristics of these braking actuators. Firstly, the total demanded braking torque can be divided into a steady part that changes less in the braking process and a dynamic part that is associated with the change of $s_s$. Secondly, the steady part is provided by the mechanical brake and the hydraulic retarder. If the steady part of demanded torque is larger than the maximum torque of the hydraulic retarder at the current speed, it will be started up and generate torque; otherwise, it will be shut down. The difference of the steady part of demanded torque and the hydraulic retarder torque will be compensated by the mechanical brake. Thirdly, the dynamic part is provided by the PMSM. Finally, the hydraulic retarder exits when the speed of the vehicle is less than 12 km/h, and the PMSM exits under the point of 5 km/h to suit its own features.

The braking torque allocation method is shown in Figure 5.

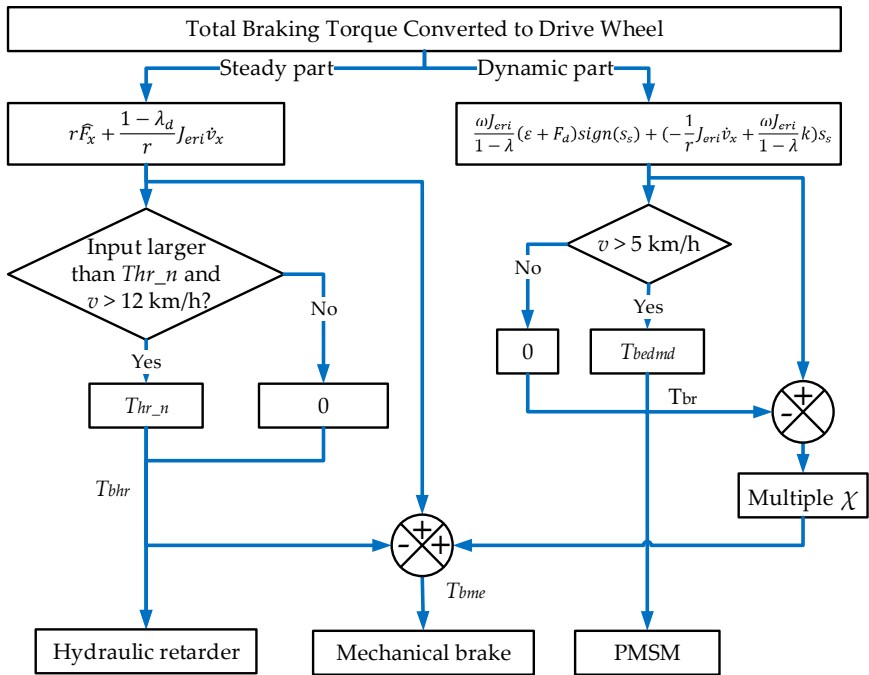

**Figure 5.** Schematic diagram of the braking torque allocation method.

Substituting the definition of the sliding surface into Equation (20), we can get that:

$$T_{bdem} = r\hat{F}_x + \frac{1-\lambda_d}{r}J_{eri}\dot{v}_x + \frac{\omega J_{eri}}{1-\lambda}(\varepsilon + F_d)sign(s_s) + \left(-\frac{1}{r}J_{eri}\dot{v}_x + \frac{\omega J_{eri}}{1-\lambda}k\right)s_s \tag{21}$$

In this equation, we can find that although $F_x$, $J_{eri}$, and $\dot{v}_x$ are functions of slip ratio $\lambda$, they change slowly when the slip ratio changes around the point $\lambda_d = 0.2$; however, the value of the last two items in this equation changes greatly with the fluctuation of $s_s$, and the value may be positive, which can only be provided by the PMSM. Thus, the steady and dynamic part of total braking torque can be written as follows:

$$\begin{cases} T_{bdems} = r\hat{F}_x + \frac{1-\lambda_d}{r}J_{eri}\dot{v}_x \\ T_{bdemd} = \frac{\omega J_{eri}}{1-\lambda}(\varepsilon + F_d)sign(s_s) + \left(-\frac{1}{r}J_{eri}\dot{v}_x + \frac{\omega J_{eri}}{1-\lambda}k\right)s_s \end{cases} \tag{22}$$

However, the dynamic part of total demanded braking torque $T_{bdemd}$ may exceed the ability of the PMSM greatly when the deviation of the real slip ratio $\lambda$ and demanded slip ratio $\lambda_d$ is large, so it is better to use the mechanical brake compensating the difference of them to improve the control effect.

Thus, the demanded torque for the mechanical brake and PMSM converted to the drive wheel can be written as:

$$
\begin{cases}
T_{bmdem} = r\hat{F}_x + \frac{1-\lambda_d}{r}J_{eri}\dot{v}_x - \hat{T}_{bhr} + \chi(T_{brdem} - T_{mo}i_ci_r) \\
T_{brdem} = \frac{\omega J_{eri}}{1-\lambda}(\varepsilon + F_d)sign(s_s) + \left(-\frac{1}{r}J_{eri}\dot{v}_x + \frac{\omega J_{eri}}{1-\lambda}k\right)s_s
\end{cases}
\tag{23}
$$

where $\hat{T}_{bhr}$ is the estimated hydraulic retarder torque converted to the drive wheel and $\chi$ is the compensating coefficient to compensate the difference of demanded dynamic braking torque and PMSM torque. The value of $\chi$ can vary from 0–1 according to the response characteristics of the mechanical brake, the hydraulic retarder, the PMSM, and the drive system to achieve a better braking performance. It is usually set with a relatively small value to prevent large overshoot of $\lambda$ due to the poor dynamic response of the mechanical brake.

To reduce chattering, the symbolic function $sign(s_s)$ can be replaced by a saturation function $sat\left(\frac{s_s}{\Phi}\right)$; thus, $T_{brdem}$ can be rewritten as:

$$
T_{brdem} = \frac{\omega J_{eri}}{1-\lambda}(\varepsilon + F_d)sat\left(\frac{s_s}{\Phi}\right) + \left(-\frac{1}{r}J_{eri}\dot{v}_x + \frac{\omega J_{eri}}{1-\lambda}k\right)s_s
\tag{24}
$$

where the saturation function $sat\left(\frac{s_s}{\Phi}\right)$ is defined as:

$$
sat\left(\frac{s_s}{\Phi}\right) = \begin{cases}
1, & s_s > \Phi \\
\frac{s_s}{\Phi}, & |s_s| < \Phi \\
-1, & s_s < -\Phi
\end{cases}
\tag{25}
$$

where $\Phi$ is a positive real number.

## 4. Simulations and Analysis

### 4.1. Simulation Setups

According to Equation (12), the highest deceleration a DDTV needs to achieve is −5.5 m/s², which means the emergency braking control strategy does not have to be triggered fundamentally when the vehicle runs on a high adhesion ground. Thus, simulations only need to be conducted under a muddy road, snow, and ice among the conditions shown in Figure 3.

Except the braking control strategy proposed in this article, the other three strategies are also taken into simulations as comparison. All four strategies are respectively:

- **FB (full braking)**: which is called "full braking" and is the most widely-used emergency braking control strategy in tracked vehicles. With this strategy, the hydraulic retarder and mechanical brake work separately to provide the largest braking torque whether the track is locked or not;

- **TABS (traditional ABS)**: which is transplanting a traditional ABS control strategy into tracked vehicles that need to add a PWM-driven solenoid valve in each wheel cylinder to realize the high-speed rise-hold-decline processes of braking pressure in it. The duty ratio of PWM is calibrated for each track-ground condition to achieve the best performance. The hydraulic retarder and mechanical brake also work separately.

- **SMSCM (sliding mode slip control without motor)**: which uses the output of the sliding mode slip ratio controller proposed in this article as the total demanded braking torque, but the PMSMs are not used in the process of emergency braking. The hydraulic retarder and mechanical brake works cooperatively to follow the demanded braking torque derived in Equation (20).

- **SMSCR (sliding mode slip control with regenerative braking)**: which is the emergency braking control strategy that the regenerative braking of PMSMs also participate in in the braking process proposed in this article.

In addition, the hydraulic retarder obeys the rule of shutting down when the vehicle runs at a speed lower than 12 km/h in all these braking control strategies above.

Three indicators are taken into the simulations to evaluate the performances of these strategies. The first one is the stopping time *T*, and the second one is the stopping distance *S*. The last indicator *D* is defined as Equation (26) to represent the deviation of the real slip ratio to the demanded value:

$$D = \left\{ \frac{1}{\lambda_d{}^2 T_q} \int_0^{T_q} [\lambda(t) - \lambda_d(t)]^2 dt \right\} \times 100\% \qquad (26)$$

where $T_q$ is the duration from when emergency braking is triggered to achieve the speed of 5 km/h such that only mechanical brakes provide the braking torque.

The simulations were performed in the environment of MATLAB/Simulink R2015b. The DDTV was braked at a speed of 80 km/h, and the brake pedal was set to increase from 0–100% with the accelerator pedal decreasing to zero in 10 milliseconds. The start point of plotting and calculating the indicators was defined as when the velocity of the track was equal to that of the vehicle to eliminate the effect of the slip of driving. The main parameters of the DDTV for the simulations are listed in Table 1.

**Table 1.** Main parameters of the DDTV for simulation.

| Parameter | Value | Parameter | Value |
|---|---|---|---|
| Vehicle mass, $m$(t) | 52 | Approach angle, $\varphi_A$ (°) | 27.3 |
| Drive wheel radius, $r$(m) | 0.309 | Departure angle, $\varphi_D$ (°) | 35.6 |
| Frontal area of the vehicle, $A$(m$^2$) | 5.36 | Ratio of the wheel-side reducer, $ir$ | 4.59 |
| Aerodynamic drag, $C_D$ | 1 | Ratio of the coupling mechanism, $ic$ | 2.2 |
| Air density, $\rho\left(Ns^2/m^4\right)$ | 1.22 | Rated power of the PMSM, $P_m$ (kW) | 625 |
| Rotational inertia of drive wheel and power train, $\left(kg{\cdot}m^2\right)$ | 158.8 | Rated speed of the PMSM, $n_m$ (rpm) | 3000 |
| Rotational inertia of idler, $\left(kg{\cdot}m^2\right)$ | 31.2 | Max speed of the PMSM, $n_{max}$ (rpm) | 9000 |
| Rotational inertia of road wheel, $\left(kg{\cdot}m^2\right)$ | 23.7 | Quantity of road wheels, half vehicle | 6 |
| Rotational inertia of support roller, $\left(kg{\cdot}m^2\right)$ | 13 | Quantity of support rollers, half vehicle | 3 |

## *4.2. Simulation Results and Analysis*

### 4.2.1. Muddy Road

A muddy road is a typical medium adhesion track-ground condition with the largest adhesion coefficient around 0.4. Simulation results of these four strategies mentioned above conducted under a muddy road are shown in Figure 6, and the value of indicators are listed in Table 2. In Figure 6, VV, TV, HRT, MBT, and MRT are used as abbreviations of vehicle velocity, track velocity, hydraulic retarder torque, mechanical brake torque and motor regenerative torque respectively.

As shown in Figure 6 and Table 2, taking any of these three strategies to prevent the locking of drive wheel in the process of emergency braking can enhance the performance greatly compared with full braking, which is the most popular way of emergency braking for tracked vehicles. The performance of SMSCR was the best among all these strategies, and it had an advantage of 0.12 s in stopping time and 1.29 m in stopping distance to the second best. As for the ability of controlling the slip ratio, the indicator D can be reduced by more than an order of magnitude with SMSCR compared with other methods. However, although SMSCM performed a little better than TABS in stopping time and controlling the slip ratio, its stopping distance was 0.79 m longer than the latter, which means that the traditional way of antilock braking performs effectively when the adhesion is not very low, and even if an advanced method is used to control the slip ratio, the performance may still not be improved significantly over the traditional way.

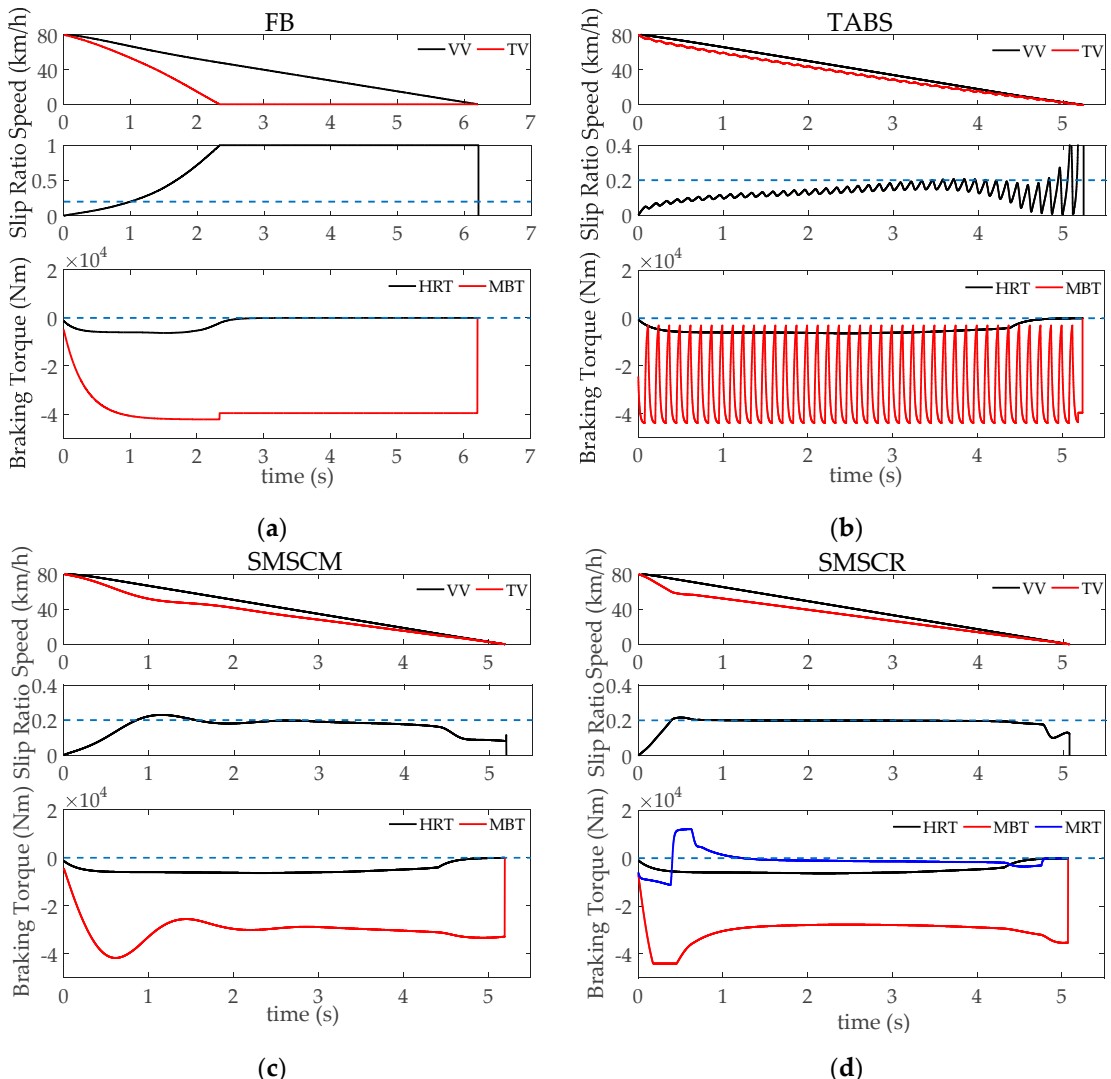

**Figure 6.** Simulation results of four emergency braking control strategies mentioned in Section 4.1 on muddy ground. They are the curves of vehicle and track speed, slip ratio and braking torque of: (**a**) full braking (FB); (**b**) traditional ABS (TABS); (**c**) sliding mode slip control without motor (SMSCM); (**d**) sliding mode slip control with regenerative braking (SMSCR). VV, TV, HRT, MBT, and MRT are vehicle velocity, track velocity, hydraulic retarder torque, mechanical brake torque, and motor regenerative torque, respectively.

**Table 2.** The value of indicators in the simulations on a muddy road.

| Indicator | FB | TABS | SMSCM | SMSCR |
|---|---|---|---|---|
| $T$ (s) | 6.21 | 5.24 | 5.19 | 5.07 |
| $S$ (m) | 67.64 | 58.45 | 59.24 | 57.16 |
| $D$ (%) | 1054.50 | 17.00 | 9.50 | 3.25 |

### 4.2.2. Snow

Snow is a typical low adhesion track-ground condition with the largest adhesion coefficient around 0.2. Simulation results of these four strategies conducted under snow are shown in Figure 7, and the values of indicators are listed in Table 3. The abbreviations are the same with braking on a muddy road.

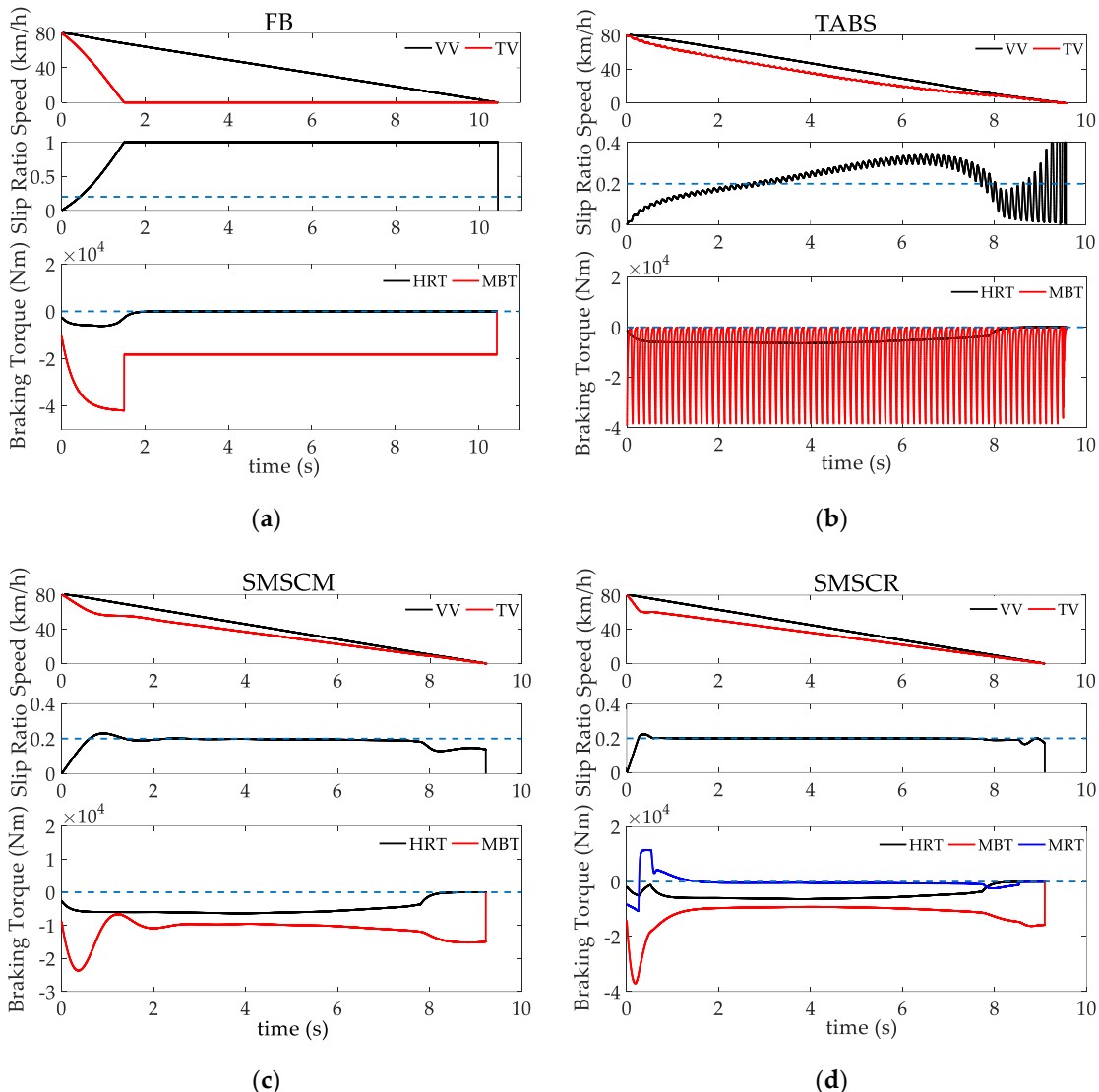

**Figure 7.** Simulation results of four emergency braking control strategies mentioned in Section 4.1 on snow. They are the curves of vehicle and track speed, slip ratio, and braking torque of: (**a**) FB; (**b**) TABS; (**c**) SMSCM; (**d**) SMSCR.

**Table 3.** The value of indicators in the simulations on snow.

| Indicator | FB | TABS | SMSCM | SMSCR |
|:---:|:---:|:---:|:---:|:---:|
| $T$ (s) | 10.75 | 9.57 | 9.23 | 9.09 |
| $S$ (m) | 115.50 | 106.15 | 103.31 | 101.74 |
| $D$ (%) | 1404.25 | 17.50 | 3.25 | 1.12 |

As shown in Figure 7 and Table 3, the last three braking control strategies still performed much better than full braking. Compared with the results of a muddy road, the deviation of the real slip ratio from the demanded value became larger when using full braking as the time of completely locking took a larger part in the whole braking process. The performances of SMSC strategies whether with the participation of regenerative braking performed by PMSMs or not exceeded the one of TABS significantly when braking on snow, while taking the PMSMs in use with the proposed allocation method remained superior to only using the mechanical brakes and hydraulic retarders. The stopping distance of SMSCR even reduced more than 1.5 m compared to SMSCM, not to mention that all indicators can be improved by more than 10% compared to full braking. What is more, we can find

that the value of all the indicators by SMSCM was improved greatly over TABS, which means with the decrease of track-ground adhesion, the control effect of the traditional antilock braking method met a change toward the negative direction, and the significance of an advanced braking control method gradually stood out.

### 4.2.3. Ice

The tracked vehicle will face an even lower adhesion coefficient when running on an icy ground than on snow. Under this track-ground condition, the largest adhesion coefficient was only a little more than 0.1. The simulation results and the value of indicators are shown in Figure 8 and Table 4. The abbreviations are the same as braking on a muddy road or snow.

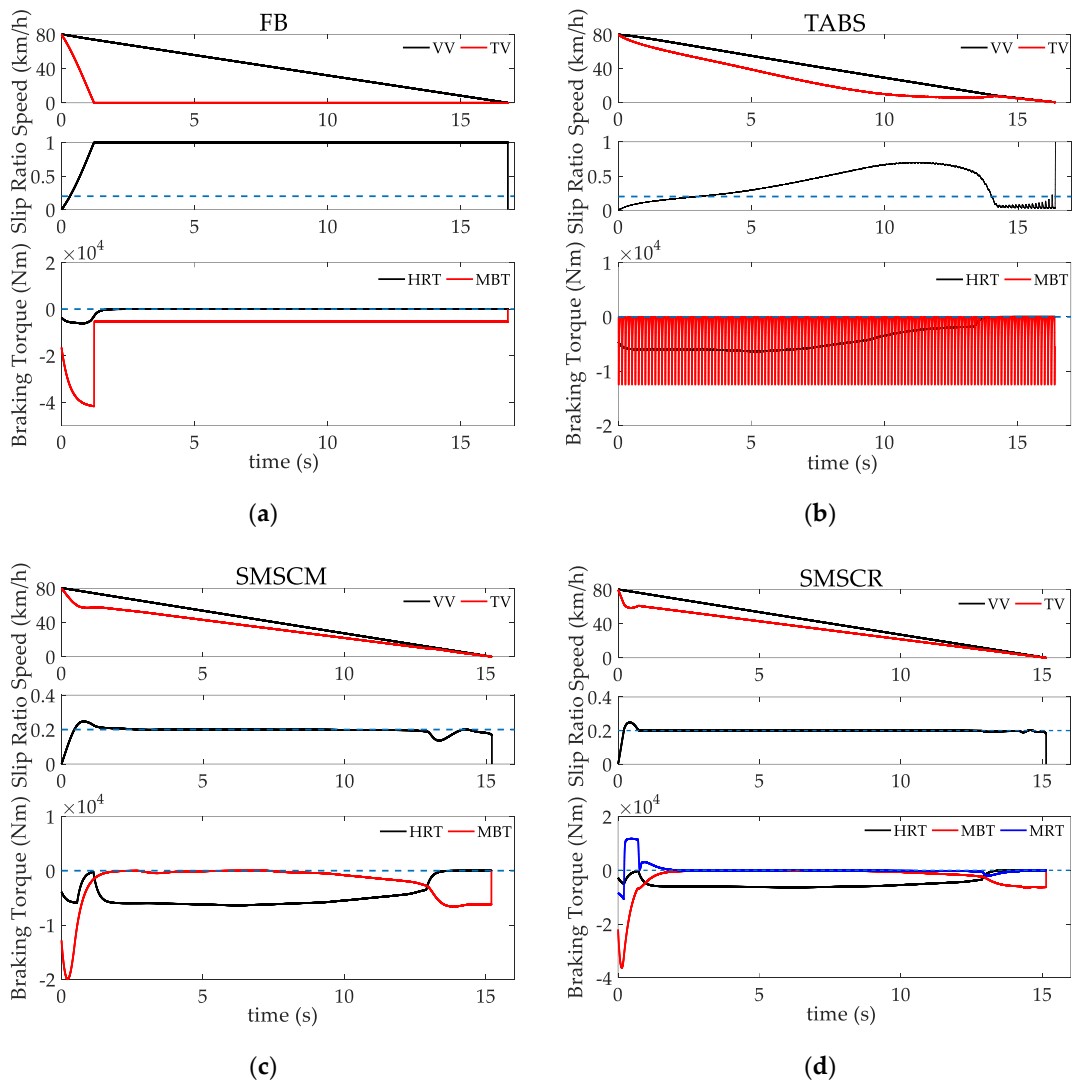

**Figure 8.** Simulation results of four emergency braking control strategies mentioned in Section 4.1 on ice. They are the curves of vehicle and track speed, slip ratio, and braking torque of: (**a**) FB; (**b**) TABS; (**c**) SMSCM; (**d**) SMSCR.

**Table 4.** The value of indicators in the simulations on ice.

| Indicator | FB | TABS | SMSCM | SMSCR |
|:---:|:---:|:---:|:---:|:---:|
| $T$ (s) | 16.77 | 16.41 | 15.20 | 15.14 |
| $S$ (m) | 185.54 | 176.50 | 168.94 | 168.20 |
| $D$ (%) | 1501.00 | 210.25 | 1.56 | 0.62 |

As we can see in Figure 3, the $\mu - \lambda$ curve of track-ice was flatter when slip ratio exceeded 0.2 than any other one shown in the same figure, which means the difference of the adhesion coefficient whether the track was locked or not was rather small, which could weaken the advantage of using an antilock braking method. Even though, the deviation of stopping distance and its percentage between these three antilock methods and full braking were still significant. However, the difference on the control effect of a vehicle using TABS and an SMSC method became larger due to the track-ground adhesion being worse.

According to Figures 6–8, we can find that there was an overshoot of slip ratio $\lambda$ at the start of braking when using an SMSC method in the process of tracking $\lambda_d$. Thanks to the ability of providing positive torques with PMSMs involved in the braking process, the convergence of $\lambda$ using SMSCR was faster than SMSCM, which led to a shorter stopping distance and better braking performance. Thus, reduced energy recovery was negligible compared to better tracking of the slip ratio.

Through the simulations above, we can conclude that although using a traditional antilock braking strategy can enhance braking performance and it even can stand in a better place compared to using an SMSC method without a motor when track-ground adhesion is rather high, the poor performance on the extremely low adhesion track-ground condition with limited maximum deceleration constrains its application. The SMSC method without a motor performed much more stable than full braking. However, its performance was worse than TABS when the track-ground adhesion was rather high, not to mention that it was always worse than the proposed strategy due to the rather poor dynamic response of the mechanical brake than PMSM and the lack of ability to produce a positive torque to accelerate the convergence of sliding mode in the process of braking. As a comparison, the SMSC method with the use of a motor proposed in this article was advantageous in whatever track-ground conditions compared to others, demonstrating a favorable value of application. As shown in Table 5, when the vehicle ran on a muddy road, snow, and ice, the stopping distance reduced by 15.49%, 11.91%, and 9.35%, respectively, and the slip ratio deviation decreased by 99.69%, 99.92%, and 99.96%, respectively. Even compared with TABS, the proposed strategy still had an advantage of reducing by 2.20%, 4.15%, and 4.70%, respectively, the stopping distance and decreasing by 80.88%, 93.62%, and 99.70%, respectively, the slip ratio deviation when running on the same ground as mentioned above, while it also performed better than the SMSC method without using the PMSM.

**Table 5.** The enhancement by using SMSCR compared with FB and TABS.

| Track-Ground Condition | FB | | TABS | |
|:---:|:---:|:---:|:---:|:---:|
| | Reduced $S$ (%) | Reduced $D$ (%) | Reduced $S$ (%) | Reduced $D$ (%) |
| Muddy road | 15.49 | 99.69 | 2.20 | 80.88 |
| Snow | 11.91 | 99.92 | 4.15 | 93.62 |
| ice | 9.35 | 99.96 | 4.70 | 99.70 |

## 5. Hardware-in-the-Loop Experiments

### 5.1. Experiment Setups

It is hard to complete the decrease of the acceleration pedal and the increase of the brake pedal in an extremely short time in a real vehicle. Communication delays of the controller with sensors and actuators, as well as differences between the PC and vehicle on-board controller may also impact the control effect. Thus, it is worthwhile to conduct some HIL experiments on the controller before applying in real vehicle. The scheme of the experiment system is shown in Figure 9.

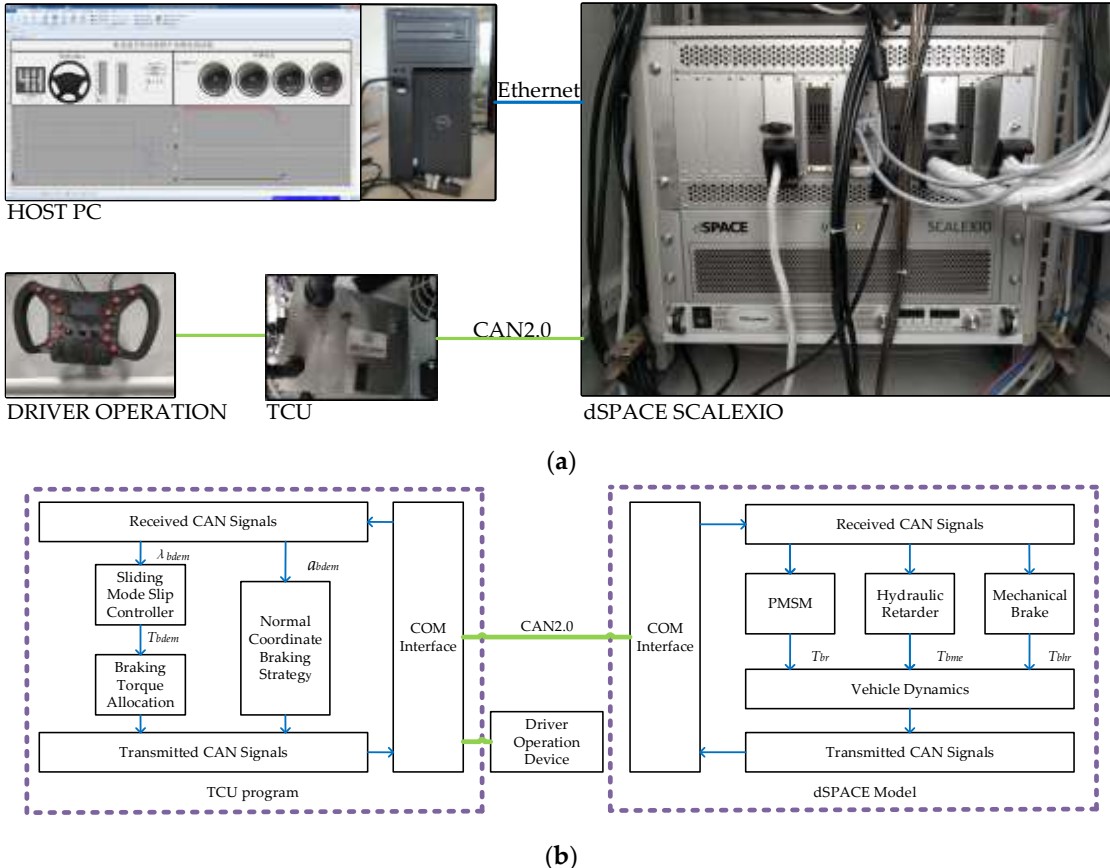

**Figure 9.** Schematic diagram of HIL experiment setups: (**a**) is the hardware level and (**b**) is the software level.

The experiment system consisted of a TCU with the proposed emergency braking control strategy, a driver input device, a dSPACE SCALEXIO HIL experiment system running the real-time model of a DDTV, and a host computer logging and plotting the data of the experiments. The TCU received control inputs from the driver input device and vehicle status signals from dSPACE, then output control instructions to dSPACE to run the real-time model. The TCU, driver input device, and dSPACE were interconnected through a high-speed CAN 2.0 network. Data transmission between dSPACE and the host PC was via Ethernet.

The experiment parameters were the same as the simulations shown in Table 1. The experiments were conducted under a muddy road, snow, and ice, respectively. The driver stepped on the accelerator pedal to the full travel until the vehicle achieved a speed of 80 km/h, then the driver released the acceleration pedal and stepped on the brake pedal to the full travel until the vehicle speed dropped to zero. The indicators proposed in Section 4 were still used to assist in analyzing the experiment results. The start points of plotting, as well as calculating the stopping time and stopping distance were defined as when the driver started to release the accelerator pedal, while the starting point of calculating indicator D was defined as when the slip ratio exceeded 0.15, which means the emergency braking control strategy was already triggered.

*5.2. Experiment Results and Analysis*

The experiment results are shown in Figure 10, and the indicators are listed in Table 6. The abbreviations are almost the same as the simulations, with the addition of Acc, Brake, TBT, and TDT, which represent acceleration pedal travel, braking pedal travel, total braking torque, and total demanded braking torque, respectively.

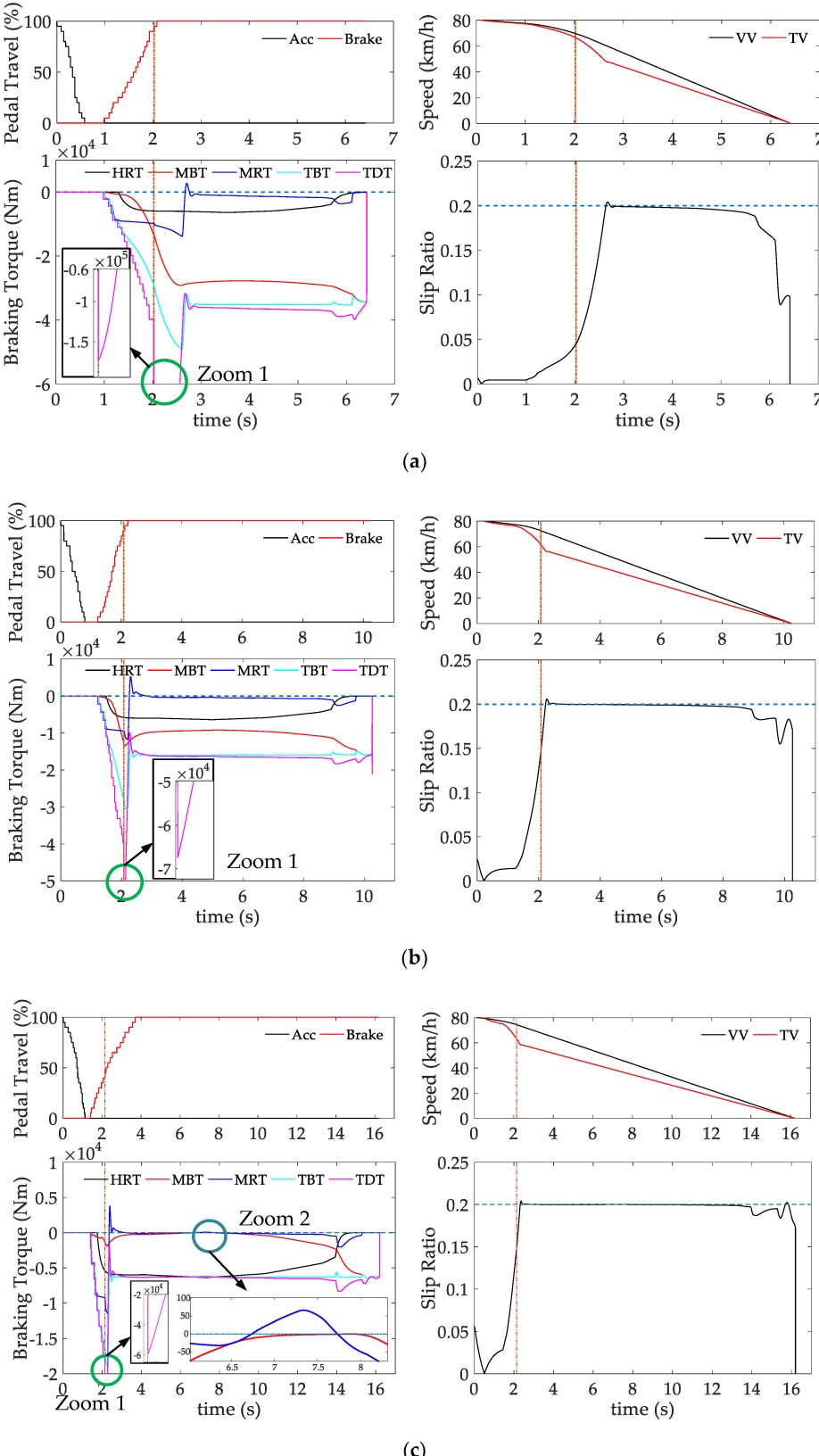

**Figure 10.** Experiment results of the proposed emergency braking control strategy under three track-ground conditions. They are respectively the curves of accelerator pedal and brake pedal travel, vehicle and track speed, braking torque, and the slip ratio of: (**a**) a muddy road; (**b**) snow; (**c**) ice. Acc, Brake, TBT, and TDT represent acceleration pedal travel, braking pedal travel, total braking torque, and total demanded braking torque, respectively.

**Table 6.** The value of indicator in the experiments.

| Indicator | Muddy Road | Snow | Ice |
|:---:|:---:|:---:|:---:|
| $T$ (s) | 6.42 | 10.26 | 16.20 |
| $S$ (m) | 85.85 | 126.94 | 191.40 |
| $D$ (%) | 0.30 | 0.11 | 0.05 |

Because of the introduction of a real driver and premeditated speed control of tapping pedals, the stopping time and stopping distance in the experiments were longer than those in the simulations to varying degrees. However, the slip ratios deviation of various track-ground conditions in the simulations and experiments were different from each other. As mentioned in Section 3, the emergency braking control strategy was triggered when the brake pedal exceeded a certain value or the slip ratio was higher than a threshold. In the simulations, the emergency braking control strategy was triggered by the brake pedal, so that the slip ratio curve had an ascent stage from zero included in the calculation of D, which made it unfair to compare with the same indicator in the experiments, which was calculated from the slip ratio rising up to 0.15; thus, the slip ratio deviation of the simulations needed to be revised. The amended slip ratio deviation $D_a$ is listed in Table 7.

**Table 7.** The value of amended slip ratio deviation in the simulations.

| Indicator | Muddy Road | Snow | Ice |
|:---:|:---:|:---:|:---:|
| $D_a$(%) | 0.14 | 0.07 | 0.13 |

As shown in Figure 10, the trigger time of emergency braking was 2.02 s, 2.09 s, and 2.15 s for a muddy road, snow, and ice, respectively. The emergency braking control strategy was triggered by brake pedal travel when braking on a muddy road, while triggered by the slip ratio when braking on snow or ice. We found that the largest deviation of total braking torque with total demanded braking torque occurred at the time when the emergency braking control strategy was triggered, shown in the Zoom 1 of all Panels (a), (b), and (c) of Figure 10, where a sharp increase of total demanded braking torque can be found. The largest braking torque demanded converted to the drive wheel when braking on a muddy road, snow, and ice were nearly 174 kNm, 68 kNm, and 60 kNm, respectively, which were far more than the braking system could provide. This phenomenon was caused by the large difference of the sliding mode $s_s$ with the desired sliding surface, and most of it was allocated to be provided by the regenerative braking of PMSM according to Section 3.2. The compensating coefficient $\chi$ was exactly designed to narrow the gap as much as possible in this situation.

Both the motor torque and slip ratio fluctuated a little more in these experiments compared to in the simulations due to the communication delay and the slower computing speed of the TCU than a PC, and we can also find in the Zoom 2 of Figure 10c that even in the flatter stage, sometimes the PMSM needed to provide a positive torque to trace the demand. The slip ratio deviation in experiment was just slightly larger than in the simulation for a muddy road, and snow and even better for ice due to the smaller overshoot caused by the brake pedal input being more gradual than in the simulations. In conclusion, the proposed emergency braking control strategy can still maintain a good control effect applied in a real TCU and actuated by real driver input.

## 6. Conclusions

In order to shorten the stopping distance while improving the maneuverability of an electric drive tracked vehicle, a novel emergency braking control strategy was presented. A sliding mode slip ratio controller was developed based on the tracked vehicle to achieve the best adhesion. Moreover, a rule-based braking torque allocating method was designed to coordinate the hydraulic retarder, the mechanical brake, and the PMSM to make the most use of their features and reduce braking torque response time. The simulation results showed that compared with full braking, the stopping

distance was narrowed, and the slip ratio deviation was apparently reduced when using the proposed emergency braking control strategy. What is more, transplanting a traditional antilock braking system would cost much in the retrofitting of the hydraulic mechanical braking circuit of a DDTV along with adding high-speed solenoid valves and their stimulators, while applying the proposed emergency braking control strategy does not require changing the original structure of the vehicle, which makes the proposed strategy achieve the best result with minimal cost. The HIL experiments verified the validity of the proposed strategy in a real TCU with real driver operations.

**Author Contributions:** Conceptualization, Z.C. and X.Z.; data curation, Z.C. and B.H.; formal analysis, Z.C.; investigation, Z.W. and Y.L.; methodology, Z.C.; project administration, Z.C. and X.Z.; resources, X.Z. and Y.L.; software, Z.C. and B.H.; supervision, X.Z.; validation, Z.C., Z.W. and Y.L.; visualization, Z.C.; writing, original draft, Z.C.; writing, review and editing, X.Z. and Z.W.

**Funding:** This research received no external funding.

**Conflicts of Interest:** The authors declare no conflict of interest.

## Appendix A

**Table A1.** Nomenclature.

| Parameter | Explanation | Parameter | Explanation |
|---|---|---|---|
| $a_{bdem}$ | demanded deceleration | $n_{rw}$ | number of road wheels |
| D | relative slip ratio deviation | $n_{sr}$ | number of support rollers |
| $F_{aero}$ | half of the air drag | $r$ | drive wheel radius |
| $F_d$ | boundary of $f$ | $r_I$ | Idler radius |
| $F_x$ | track-ground resultant force | $r_{rw}$ | road wheel radius |
| $\hat{F}_x$ | estimation of $F_x$ | $r_{sr}$ | support roller radius |
| $\widetilde{F}_{x,max}$ | the maximum error of estimating $F_x$ | $s$ | Laplace's variable |
| $F_t$ | track force | $s_s$ | sliding mode |
| $i_c$ | transmission ratio of the coupling mechanism | S | stopping distance |
| $i_r$ | transmission ratio of the wheel-side reducer | $sat$ | saturation function |
| $J$ | rotational inertia of drive wheel and transmission system | $sign$ | sign function |
| $J_{eri}^{I}$ | equivalent rotational inertia of the road wheel | $v_x$ | longitudinal velocity |
| $J_I$ | rotational inertia of the road wheel | T | stopping time |
| $J_{eri}^{rw}$ | equivalent rotational inertia of the road wheel | $T_b$ | braking torque applied on the drive wheel |
| $J_{rw}$ | rotational inertia of the road wheel | $T_{mo}$ | motor torque |
| $J_{eri}^{sr}$ | equivalent rotational inertia of the support roller | $T_{brdem}$ | demanded motor torque converted to the drive wheel |
| $J_{sr}$ | rotational inertia of the support roller | $T_{mo\_max\_n}$ | maximum torque the motor can output at a speed of n |
| $J_{eri}^{t\_f}$ | equivalent rotational inertia of the front part of the track | $T_{me}$ | torque of the mechanical brake |
| $J_{eri}^{t\_r}$ | equivalent rotational inertia of the rear part of the track | $T_{bmdem}$ | demanded torque of mechanical braking converted to the drive wheel |
| $J_{eri}^{t\_u}$ | equivalent rotational inertia of the upper part of the track | $T_{hr}$ | real torque of the hydraulic retarder |
| $J_{eri}^{t\_b}$ | equivalent rotational inertia of the bottom part of the track | $T_{hr\_n}$ | the torque the hydraulic retarder can output at a speed of n |
| $J_{eri}$ | equivalent rotational inertia | $T_{bhr}$ | the torque of the hydraulic retarder converted to the drive wheel |
| $M$ | half mass of the vehicle body | $\beta$ | brake pedal travel |
| $m_{t\_f}$ | the mass of the front part of the track | $\lambda$ | slip ratio |
| $m_{t\_r}$ | the mass of the rear part of the track | $\lambda_d$ | demanded slip ratio |
| $m_{t\_u}$ | the mass of the upper part of the track | $\mu$ | adhesion coefficient |
| $m_{t\_b}$ | the mass of the bottom part of the track | $\omega$ | Rotational speed of the drive wheel |

## Appendix B

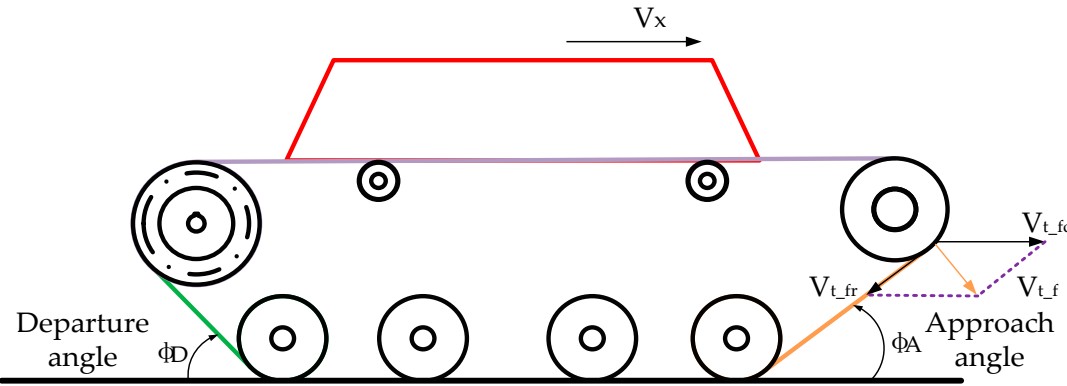

**Figure A1.** Schematic diagram of the velocity resolution of a DDTV.

Since the calculating of ERI is based on energy conservation theory, the key to amending Equation (5) into Equation (7) is calculating the speed of each part of the track with the slip ratio $\lambda$ taken into consideration when braking. Taking the front part of the track as an example, the speed $v_{t\_f}$ can be decomposed into a converted velocity $v_{t\_fc}$, which equals the speed of the vehicle, and a relative velocity $v_{t\_fr}$, which equals the linear velocity of the wheel rotation, as shown in Figure A1. Thus, taking Equation (6) into consideration, the dynamics of the front part of the track can be expressed as:

$$\begin{cases} \overrightarrow{v_{t\_f}} = \overrightarrow{v_{t\_fc}} + \overrightarrow{v_{t\_fr}} \\ v_{t\_fc} = v_x = \frac{\omega r}{1-\lambda} \\ v_{t\_fr} = \omega r \\ \frac{1}{2}m_{t\_f}v_{t\_f}^2 = \frac{1}{2}J_{eri}^{t\_f}\omega^2 \end{cases} \tag{A1}$$

According to the cosine law, $v_{t\_f}^2$ can be expressed as:

$$v_{t\_f}^2 = v_{t\_fc}^2 + v_{t\_fr}^2 - 2v_{t\_fc}v_{t\_fr}\cos\varphi_A \tag{A2}$$

Substituting $v_{t\_f}$ as a function of $v_{t\_fc}$, $v_{t\_fr}$, and $\varphi_A$ from Equation (A2) into (A1), the ERI of the front part of the track can be derived as:

$$J_{eri}^{t\_f} = m_{t\_f}r^2\left[1 + \frac{1}{(1-\lambda)}^2 - \frac{2\cos\varphi_A}{(1-\lambda)}\right] \tag{B3}$$

which is exactly the same as what is written in Equation (7).

The expression of other three parts of the track can be derived using the same way above.

## Appendix C

The derivation of Equation (20) is mainly based on the theories described in [38]. From Equation (19), we can design:

$$T_{bdem} = \frac{\omega J_{eri}}{1-\lambda}\hat{f} + \frac{\omega J_{eri}}{1-\lambda}[\varepsilon sign(s_s) + ks_s] - \frac{\omega J_{eri}}{1-\lambda}f_c \tag{A3}$$

where $f_c$ is a positive real number under design, which is related to the boundary of $\widetilde{f}$.

Substituting $T_b$ defined in Equation (A1) into Equation (19), we can get:

$$\dot{s}_s = -\varepsilon sign(s_s) - ks_s + f_c - \widetilde{f} \tag{A4}$$

From Equation (17), we can find that the boundary of $\widetilde{f}$ is:

$$-F_d \leq \widetilde{f} \leq F_d \tag{A5}$$

Thus, to ease the sliding mode arrival condition, $f_c$ can be designed as:

$$f_c = \frac{F_d + (-F_d)}{2} - \frac{F_d - (-F_d)}{2} sign(s_s) = -F_d sign(s_s) \tag{A6}$$

Substituting $f_c$ designed in Equation (A4), then we can get the expression of Equation (20).

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
