# Peer review of "A Novel Emergency Braking Control Strategy for Dual-Motor Electric Drive Tracked Vehicles Based on Regenerative Braking"

_applsci, doi:10.3390/app9122480_

Round 1
Reviewer 1 Report
Likely, a nomenclature section would be required.
Author Response
Dear reviewer:
Thank you very much for your attention and precious time on our paper “A Novel Emergency Braking Control Strategy for Dual-Motor Electric Drive Tracked Vehicles based on Regenerative Braking”. Your professional comments and suggestions are really very helpful for revising and improving our paper, as well as the important guiding significance to our future researches. We have studied these comments carefully and have revised the manuscript according to your kind advices which we hope will meet with approval. Thank you very much for all your attention on our paper. The responses to your comments are as following "PDF" file.

Reviewer 2 Report
The topic of the paper is quite interesting, as well as the proposed emergency braking strategy. However, many important issues arise throughout the paper, which must be addressed properly, as detailed in the following:
- please cite some references for supporting the statements reported in the first paragraph of the Introduction (use of dual-motor electric drive in tracked vehicles); in addition, please also explain more clearly which are the benefits of DDTV (two independent motors? Electric motors with fast dynamic response?);
- the operating principle of the coupling mechanism (Section 2.1, page 3, lines 97-100) is not clear, especially its aim, please explain it better by enclosing additional details;
- considering Fig. 1, it seems that GCU and MCU should be DC/AC and AC/DC converters respectively, together with their control units; therefore, please redraw Fig. 1 by highlighting each of these components (converter and control unit) for both GCU and MCU; please also replace “DC/DC convertor” with “DC/DC converter”;
- (1) is unconvincing for two main reasons: first, traction effort and motor torque seems missing in the first and second expressions respectively. Second, if traction effort is enclosed in Fx, its direction should be reversed in Fig. 1. Please check this equation carefully;
- (5) should be moved after part of the following paragraph (page 4, lines 138-141). Furthermore, a figure is needed that shows how the vehicle is split into four parts and which are the rotational axis considered for computing the all the inertia coefficients;
- the genesis of (7) should be justified further, especially how it derives from (5) and (6);
- Section 2.4, page 6, line 165, please clarify what is intended for “PMSM external characteristic”;
- Section 3, page 7, line 200, please clarify what is intended for “certain point”;
- considering (18) and the subsequent equations, please use another variable instead of “s”, which is already employed for denoting the Laplace’s variable;
- considering (19), stating that “k” should be relatively large and “ε” relatively small is weak informative; please quantify how much these values should be large and small (compared to what?);
- the genesis of (20) is unclear, please better explain how it is derived from (15), (17) and (19);
- Fig. 5 and the corresponding explanation reported in the main text (Section 3.2) is not clear; please redraw the flowchart by highlighting how the total braking torque is split into steady-state and dynamic components at first (by introducing (21) and (22) I think). Then, operating principle and yes/no conditions of each part of the flowchart should be better highlighted and explained, by referring to equations when needed;
- Section 3.2, page 9, lines 246-247, please justify why the PMSM torque control system gets worse at low speed operation. Moreover, considering page 10, lines 267-268, please explain what is the meaning of the compensating coefficient “X”;
- Section 4.1, please explain the role of the solenoid valve used in TABS with few more details;
- Section 4.2, Figs. 6d, 7d and 8d are not fully convincing because the PMSM torque is largely positive (accelerating torque) at the start of the braking process: such an anomalous evolution must be justified properly, because it seems impairing regenerative braking capability of the PMSM. In this regard, the overall amount of energy recovered by the PMSM during braking should be disclosed for each simulation case;
- considering Fig. 10, the zoomed sight reported in the pictures “2-1” (second row, first column) is not clear, please state what it should highlight in the main text. In addition, the evolution of PMSM torque there reported is totally unacceptable because a too much larger ripple occurs; since the authors state that this problem is due to communication and/or computational effort, it must be solved appropriately and HIL experiments must be repeated accordingly;
- Section 5.2, page 17, lines 412-416, please clarify the reason why D should be amended.
Please address also the following minor issues:
- please define DDTV at the start of the introduction (page 1, line 32), although it is already defined in the abstract; the same goes for PMSM used on page 5, line 157;
- replace “electrization” with “electrification” throughout the paper;
- Section 1, page 2, line 49, the meaning of “patrol vehicle” is not clear, please check if a more appropriate term should be used;
- considering Fig. 2, it seems that “ωr” should be simply “ω” in order to make this symbol consistent with that used in the equations;
- define “r” used in (2)-(4);
- Section 2.2, page 5, line 147, rephrase the sentence there reported by stating that Jeri is achieved by summing all the inertia coefficients reported in (2)-(4) and (7);
- please clarify that “s” used in (9) and in the subsequent equations is the Laplace’s variable;
- increase the graphical resolution of Figs. 3, 9, and 10;
- Section 3.1, page 7, line 209, it seems that “substitute λ defined in equation (6)” should be replaced by “substitute ω as a function of λ from equation (6)”;
- Section 4.1, please define the acronym of each control strategy there reported;
- considering (26), it seems that “λd” should be squared in the denominator outside the integral;
- replace “chapter” with “section” in the caption of Figs. 6-8 and throughout the main text of Section 5;
- Section 6, please remove all the numeric values reported on lines 446-448, which could be eventually moved in a table to be enclosed in the previous section;
- revise the paper carefully by fixing some typos, namely by replacing “the best in all strategies” with “the best among all strategies” (abstract), “analyze” with “analysis” (Section 1, page 2, line 58), “tradition” with “traditional” (Section 1, page 2, line 70), “convertor” with “converter (Section 2, page 3, line 105), “rare” with “rear” (Section 2, page 4, line 138), “therefor” with “therefore” (Section 2, page 6, line 182), “symbolic” with “sign” (Section 3, page 8, line 228) and so on.
Author Response
Dear reviewer:
Thank you very much for your attention and precious time on our paper “A Novel Emergency Braking Control Strategy for Dual-Motor Electric Drive Tracked Vehicles based on Regenerative Braking”. Your professional comments and suggestions are really very helpful for revising and improving our paper, as well as the important guiding significance to our future researches. We have studied these comments carefully and have revised the manuscript according to your kind advices which we hope will meet with approval. Thank you very much for all your attention on our paper. The responses to your comments are as follows in the “PDF” file.

Reviewer 3 Report
- The main contribution should be compared to the previous methods. - How was the slip ratio reference obtained? - Why was the sliding mode control used for the slip control? - The qualities of Figure should be improved. - The detailed block diagram of the HILS should be added.Author Response
Dear reviewer:
Thank you very much for your attention and precious time on our paper “A Novel Emergency Braking Control Strategy for Dual-Motor Electric Drive Tracked Vehicles based on Regenerative Braking”. Your professional comments and suggestions are really very helpful for revising and improving our paper, as well as the important guiding significance to our future researches. We have studied these comments carefully and have revised the manuscript according to your kind advices which we hope will meet with approval. Thank you very much for all your attention on our paper. The responses to your comments are as follows in the “PDF” file.

Round 2
Reviewer 2 Report
The authors have improved the paper significantly, by following most of my suggestions. However, some issues are still pending, as detailed below:
- regarding (1) and Fig. 1, it is necessary to specify that they refer to vehicle braking only;
- although 600 dpi is surely a high enough graphical resolution, Figs. 3 and 9 appear a little bit fuzzy; in this regard, please check if any resolution downgrade occurs when converting the source file into PDF;
- line 173, I suggest using “torque-speed curve” rather than “external characteristics”;
- lines 203 and 208, I suggest replacing “certain point” with “threshold” or something similar in order to avoid misunderstandings;
- line 280, please explain why dynamic torque that cannot be fulfilled by PMSM cannot be provided by mechanical brake directly but it is necessary to use a compensating coefficient X; in addition, this coefficient must be explained in this section (does it varies from 0 to 1 or it is set at a constant value? Which value and how choosing it?);
- considering (26), if D would represent the variance of λ, “λd” must be removed from the denominator outside the integral, together with “x100%” at the end of the equation. Otherwise, “λd” must be squared in the denominator in order to make D consistent with a per unit variable;
- Section 4.2, Figs. 6d, 7d and 8d, in my previous comment I was not questioning the fact that PMSM torque can be positive during braking, but what I would like to underline is that the reader expects that PMSM contributes to braking by delivering a negative torque, leading to regenerative braking. However, this does not occur completely in the proposed braking strategy, so it is necessary to discuss in Section 4 why DDVT braking benefits from positive PMSM torque. In addition, if recovered energy is negligible, please state this in the main text, still in Section 4;
- Fig. 10a, picture on the second row-first column, the zoomed sight there reported seems “completing” the graphs as y-axis seems too short; if so, please increase negative range of y-axis. The same goes also for all the other figures (Figs. 10b, 10c and 10d);
- line 498, “(A2) into (A1)” should be replaced with “(B2) into (B1)”.
Author Response
Dear reviewer:
Thank you deeply for your attention and precious time on our paper “A Novel Emergency Braking Control Strategy for Dual-Motor Electric Drive Tracked Vehicles based on Regenerative Braking”. Your two round professional comments and suggestions are really very helpful for revising and improving our paper, as well as the important guiding significance to our future researches. We have studied these comments carefully and have revised the manuscript according to your kind advices which we hope will meet with approval. The responds to your comments in the second round are as following “PDF” file.

Round 3
Reviewer 2 Report
The authors have addressed all the remaining issues successfully, just the following few minor improvements should be introduced.
- Figs. 3 and 9b still appear a little bit fuzzy, please fix this;
- line 387, please rephrase the sentence “Comparing with that, the cost of decreasing regenerated energy is worth.” in order to make its meaning clearer (reduced energy recovery is negligible compared to better tracking of the slip-ratio);
- Fig. 10, please make y-axis range of the zoomed sights contiguous with that of the main figures.
Author Response
Dear reviewer:
Thank you deeply for your attention and precious time on our paper “A Novel Emergency Braking Control Strategy for Dual-Motor Electric Drive Tracked Vehicles based on Regenerative Braking”. Your three rounds professional comments and suggestions are really very helpful for revising and improving our paper, as well as the important guiding significance to our future researches. We have studied these comments carefully and have revised the manuscript according to your kind advices which we hope will meet with approval. The responds to your comments in the second round are as following “PDF” file.
